# Design Thinking for Public R&D: Focus on R&D Performance at Public Research Institutes

Seonyeong Lim [1], Minseo Kim [2] and Yeong-wha Sawng [1,*]

1 Department of Management of Technology, Konkuk University, Seoul 143701, Korea; asdv728@konkuk.ac.kr
2 College of General Education, Seoul Women's University, Seoul 01797, Korea; mskim@swu.ac.kr
* Correspondence: sawng@konkuk.ac.kr

**Abstract:** Korean public research institutes (PRIs) have shown noteworthy technological innovation over the past years, but they have been lagging in the commercialization of technological results. To ensure sustainable technology commercialization, not only technological innovation but also a shift towards a market-oriented approach is required. As such, it has become even more important for the public sector to have a demand-oriented approach for responding to market failures or demand issues in the private sector. Yet the public sector has been geared to a supply-oriented approach, while adopting a demand-oriented perspective, in its inception. In an ever-increasingly complex society, the process of Design Thinking is necessary in the both the public and private sectors. However, Design Thinking-related studies have concentrated on the latter. Studying the impacts of Design Thinking as a demand-based innovation methodology of public institutions, this research aims to expand the traditional scope of the study of Design Thinking to include PRIs—owing to relevant key research experts, PRIs will prompt changes in the overall public sector going forward. With the Design Thinking process requiring empathy, integrative thinking, and experimentalism, this study examines the impacts of Design Thinking on PRIs. This research also aims to demonstrate that Design Thinking boosts innovation, specifically through PRIs, promoting higher discourse on Design Thinking. Concluding that Design Thinking improves technology performance in public research institutes, the study evaluates that Design Thinking leads to research innovation in a demand-driven R&D environment, producing innovations in the overall public sector.

**Keywords:** design thinking; R&D performance; public R&D; public research institutes

## 1. Introduction

Studies on the differences between the private and public sectors have long fueled academic debates [1]. In general, private sector entities (both individuals and businesses) in a capitalist market have freely engaged in economic activities, and the market has allocated resources most times in a rational way. Nonetheless, whenever the market does fail to distribute resources in a rational way, the government has stepped in by providing public policies, public R&D activities, and public services. Based on this, it can be said that the public sector needs to conduct R&D activities in a way that deals with failures or demand issues in the private market, thereby making it necessary to develop policies and services based on a demand-driven approach. Over the recent years, a demand-oriented perspective has been emphasized in the public sector, as evidenced by the adoption of demand-driven R&D activities, policies, and public services [2,3]. This is a huge shift from the past, when public sector entities focused on policies and services from the perspective of suppliers, or the government. The private sector has long conducted studies to better meet the requirements of the demand side [3]. However, the public sector has been relatively slower in realizing the importance of a demand-oriented viewpoint—a development that explains why the public sector's innovation-based performance has been lackluster despite long-lasting ongoing efforts to innovate policies and services [3]. With modern society growing

increasingly complex, calls for innovations based on a demand-oriented perspective have been rising in the public sector, compelling public sector entities to solve problems with innovative solutions based on improved understanding of the requirements of the demand side. Therefore, in-depth research on consumer-centered demand-based innovation in the public sector is needed. Among the many areas in the public sector, we concentrate on the public R&D field, as output and outcomes related to scientific technologies based on public R&D activities have a significant impact not only on those having a demand for such R&D achievements, but also on organizational capabilities, policies, and environments of the overall public sector [4]. That is, demand-driven R&D activities would set the framework for creating demand-oriented policies and services. Therefore, this study looks at the demand-driven R&D of public research institutes, which play vital roles in the public R&D arena.

Representing a key pillar in public R&D and national innovation systems, public research institutes have contributed to improving scientific technologies in Korea by undertaking effective and efficient R&D programs [5–8]. They have also striven to promote growth in private sector R&D by providing R&D expertise and manpower and developing an ecosystem for industrial and academic research. Coincidently, with the private sector's enhancing R&D capabilities, public research institutes are increasingly required to assume new roles [9]. In fact, it is becoming increasingly necessary for public R&D professionals to conduct R&D activities from a demand-oriented perspective in relation to current social issues and agendas (i.e., COVID-19 and ESG) [9,10]. As such, it can be said that one of the reasons that has raised the need for demand-driven R&D is the shift in R&D environments (entailing those in related paradigm, policies, and strategies) owing to socioeconomic changes. In phase with shifting R&D environments including the emergence of the fourth-generation R&D model [2,7,11–13], R&D policies for innovative technologies [2,14–16], and direct commercialization strategies [12,17–24], it has become even more important for public research institutes to conduct demand-driven R&D. Still, examining the R&D performance of public research institutes of Korea over recent years, it appears that demand-driven R&D has yet to take hold. (Public research institutes in Korea have shown an annual average growth rate of 5.4% in technology commercialization over the past five years, yet such growth does not seem to be high enough when accounting for their R&D investment success rate of 98% [19]. Furthermore, the licensing fee per technology transfer has declined at an annual average rate of 0.2% over the past five years, which suggests a slowdown from a qualitative approach [20].) This is attributed to the fact that a lack of understanding of the requirements of the demand side has led to a divide between development and commercialization, leading to a failure in commercialization [21]. As such, in order to bridge the divide between development and commercialization and achieve successful commercialization, there should be active communication and partnership between technology suppliers and those having a demand for technology as well as an understanding of the requirements of the demand side and related stakeholders [21–24]. In the end, not only companies in the private sector, but also public research institutes in the public sector need to adopt a demand-driven approach and develop measures to understand the requirements of the demand side in the early stages of R&D.

As part of efforts to better understand the demand side, the private sector has focused on Design Thinking. (A methodology for successful innovation based on a repeatable, humanistic design method that considers consumer needs from the early phase of R&D projects [19]. In this study, Design Thinking refers to an approach to problem-solving based on humanistic observations, empathy, and integrative thinking.) Various researchers and institutions across the private sector have discussed Design Thinking. In fact, a number of precedent studies (emphasizing those in business administration departments) have demonstrated that Design Thinking contributes to corporate innovation when integrated into business strategies or models, which have prompted many companies to adopt the process of Design Thinking.

It is believed that an introduction of Design Thinking in public would help public research institutes boost demand-driven R&D, which has grown important over the recent years amid changing R&D environments. Public sector entities have recently begun to adopt Design Thinking, as evidenced by the introduction of training programs for cultivating design-thinking capabilities, living labs, and the People Make Policy initiative. However, despite such efforts in the public sector, academic research on the achievements of Design Thinking is still focused on the private sector, such as those in regard to college programs and corporate management activities [24–33]. Given this situation, rather than focusing on the private sector, which has long been the subject of Design Thinking-related studies, this study aims to look at how the adoption of Design Thinking as a methodology for technological innovation in the public R&D arena would affect technology performance, while suggesting implications for improving R&D efficiency and performance. Accordingly, the research has set public research institutes, which play the primary roles in the public R&D field, as focused research actors.

Furthermore, precedent empirical studies were mostly based on simple/multiple regression analyses that took Design Thinking as an independent variable, with their results varying by sample, thereby representing a challenge to generalizing the correlation between Design Thinking and performance. This study aims to supplement the limits of precedent studies by building up hypotheses in regard to the correlation between Design Thinking and performance based on a program logic model.

Notably, this research will be the first to demonstrate the effects of Design Thinking as a process adopted at public research institutes, which play primary roles in the public R&D sector, amid rising calls for a demand-driven approach in the public sector, and its results will likely be utilized for establishing public R&D strategies and policies going forward.

## 2. Theoretical Background

### 2.1. Design Thinking

Design Thinking is a methodology that attempts to solve problems in an innovative and human-centered way. The idea of Design Thinking can be traced back to Herbert Simon and his book "Design as a Way of Thinking" written in 1969. In the early days, Design Thinking was focused on research and education on designers' cognitive activities [24–28]. As time elapsed, however, the idea of Design Thinking came to be embraced by a number of other fields, including engineering, education, business administration, and social studies [28]. In 1991, David Kelley, the founder of IDEO (A leading design and consulting firm, established in 1991, that has promoted Design Thinking worldwide) introduced Design Thinking as a business innovation methodology for companies, paving the way for the spread of Design Thinking in business administration studies.

Design Thinking is still in its formative years and being applied to various areas, thereby making it difficult to set limits to its scope. Furthermore, its definition varies by individual researcher. One researcher once defined Design Thinking as a practical way of thinking for innovation [34]. Practical thinking here refers to a problem-solving process where the relation between a problem and related factors is identified based on empathetic interaction and undergoes reorganization [34]. Another researcher described Design Thinking as a problem-solving methodology for discovering potential customer needs by analyzing customer empathy and coming up with possible solutions [35]. In a broad sense, Design Thinking is defined as a creative thinking process for solving problems in human life in an innovative way using integrative thinking skills [22]. From the perspective of R&D, Design Thinking is defined as a methodology that drives successful innovation by factoring user needs from the early stages of R&D activities through a design-based approach (undertaking repeated experiments with a human-centered approach) [29].

Overall, this study defines Design Thinking as an innovation methodology that promotes the success of innovative efforts through discovery of explicit/potential customer needs based on human-centered empathy and observation, integrative thinking, and experiments.

## 2.2. Design Thinking Elements

"Design Thinking" has garnered significant attention with business media outlets and has been heralded as a novel problem-solving methodology well suited to the often-cited challenges business organizations face in encouraging innovation and growth [36].

As Design Thinking emerges on the cusp of broad business adoption, four critical research questions surface: (1) What is Design Thinking?; (2) How does it work?; (3) Does it lead to innovation success?; (4) If so, under which circumstances? [32]. Though writings on Design Thinking are rapidly increasing, they are largely anecdotal or prescriptive in nature, and short on rigorous, research-based insights [36,37].

While the precise terminologies describing the formal methods used in Design Thinking can differ by each author, formal methods that underlie a Design Thinking approach are established, with common themes emerging.

Table 1 is a summary of the concepts proposed by each research institute. A review of them reveals a widely shared view of the design-thinking process, despite each using different terminologies [36]. Furthermore, the concepts appear similar to the steps in the technological innovation process—exploring ideas for discovering latent market demand, diagnosing problems based on the identified latent demand and finding solutions, as well as testing/executing selected alternatives.

**Table 1.** Models of Design Thinking Process in Practice.

| Stage | IDEO | Rotman Business School | Darden Business School |
|---|---|---|---|
| Stage 1 data gathering about user needs | Discovery and interpretation | Empathy | What is? |
| Stage 2 idea generation | Ideation | Ideation | What if? |
| Stage 3 testing | Experimentation and evolution | Prototyping and experimentation | What wows? What works? |

Design Thinking is presented as either a mindset or course of action. Studies viewing Design Thinking as a mindset are based on innovative studies noting that specific ways of thinking affect one's actions or on arguments that underlying beliefs of individual abilities influence decision making, an idea promoted in implicit theories of intelligence [38]. In terms of Design Thinking actions, the literature points to diverse activities. These activities collectively form a progression of steps iteratively taken to develop an invention. Meanwhile, concepts presented in Table 1 are based on a view that regards Design Thinking as a set of activities [32]. Such activities form the procedures that are undertaken repetitively to collectively develop an invention. In this regard, Nakata and Hwang pointed out that the three activities shown in Table 1 constitute the fundamental pillars. Major attributes of design thinkers include empathy, integrative thinking, optimism, experimentalism, and collaboration [39]. With Design Thinking being considered a tool for execution, it can be said that the key attributes of design thinkers reflect those of a Design Thinking process, while learning them should help us understand the essence of Design Thinking and establish a process for generating results [29].

Based on the results of precedent studies, this research applies the characteristics of Design Thinking in three stages as follows. Stage 1 describes an empathy-based understanding of market demand, and thus can be viewed as a concept related to the perspective of Design Thinking as a mindset. Stages 2 and 3 are steps in which new solutions are tested to solve a problem based on an understanding of market demand, and represent a behavioral perspective.

## 2.3. The Characteristics of Public Sector and Design Thinking

Until now, empirical studies on Design Thinking have analyzed its efficacy, centering on individuals and businesses in the private sector. Even though the public sector has

recently joined the move to adopt Design Thinking by applying it in public policies and services, related empirical studies have been lacking. Notably, research for Design Thinking in the R&D arena has been concentrated on the private sector, analyzing its impacts on corporate performance. However, it is noteworthy that both private and public R&D differ from each other in terms of the key objectives for R&D activities and performance measurement. As shown Table 2, The optimal form for Design Thinking in an organization depends on its purpose [40].

**Table 2.** Comparison of Characteristics of Private and Public Sectors.

|  | **Private Sectors [41]** | **Public Sectors [41]** |
|---|---|---|
| Basic objective | Earning profits | Pursuing public good |
| Organizational culture | Horizontal | Vertical |
| R&D objective | Gaining a competitive advantage, creating profits | Filling technological gaps, solving social problems |
| Research results | High level of appropriation | Strong external effect |

Private sector R&D professionals seek to secure product development capabilities in order to improve their competitiveness, accumulate knowledge, and foster talent for profit making. Various researchers and institutions across the private sector have discussed Design Thinking. According to the results of precedent studies, Design Thinking is a methodology that helps people and organizations not only to pursue innovation, but also to solve problems in the private sector by enabling them to design more effective and efficient solutions based on innovative and creative thinking skills [24–28]. In fact, a number of precedent studies (emphasizing those in business administration departments) have demonstrated that Design Thinking contributes to corporate innovations when integrated into business strategies or models, which have prompted many companies to adopt the process of Design Thinking.

While public sector entities, especially public research institutes (public research institutes refer to institutions that conduct state-led R&D projects for scientific technologies [5]), have many different roles, they have put significant emphasis on improving private R&D performance by providing R&D expertise and manpower, and by developing an ecosystem for industrial and academic research. Coincidently, with the private sector's enhancing R&D capabilities, public research institutes are increasingly required to assume new roles [9]. In fact, it is becoming increasingly necessary for public research institutes to conduct R&D activities in relation to current social issues and agendas (i.e., COVID-19 and ESG). That is, public R&D professionals should not only contribute to industrial and economic growth, but should also undertake demand-driven R&D activities aimed at improving people's quality of life by addressing social issues [10]. As such, it can be said that one of the reasons that has raised the need for demand-driven R&D in public research institutes is the shift in R&D environments (entailing those in related paradigm, policies, and strategies) owing to socioeconomic changes.

Notably, fourth-generation R&D is a new R&D model that has emerged amid the advance of social and economic systems, increased complexity of social trends and affairs, and accelerating technological innovation. Going beyond a supply-oriented perspective, fourth-generation R&D puts significant emphasis on developing new innovation systems and platforms by fostering partnerships with various networks [2]. That is, it emphasizes the reflection of both explicit and potential needs not only of direct users, but also other stakeholders (including potential users) by enabling close partnerships with them [7]. To that end, it is becoming increasingly important to clearly define the scope of stakeholders and engage in close partnerships with them throughout the whole R&D process [2,12,13].

With changes in the R&D model leading to new R&D policies, such policies have shifted from a fast-follower strategy to a first-mover one. A fast-follower approach is

useful in that it creates high added value and facilitates fast economic growth by adopting advanced overseas technologies [2,14,15]. Meanwhile, as a strategy that increases dependency on other countries, the approach does not fit with the fourth-generation R&D model, which emphasizes that public entities address social issues and pursue innovation [2,14,15]. Instead, a first-mover strategy appears more suitable for the fourth-generation R&D model since it emphasizes a researcher's autonomy, creativity, experimentalism, and efforts for convergence, and prioritizes qualitative growth in R&D [16]. Accordingly, countries around the world have been shifting their R&D policies from a fast-follower approach (which pursues economic efficiency by obtaining technologies from abroad) to a first-mover one (which focuses on human-centered solutions for social issues) [16], while public research institutes have put great effort into building a trust-based system that promotes a spirit of expertise and challenge by embracing failure [16].

Furthermore, there have been changes in strategies to commercialize R&D results. In the past, public research institutes pursued a conservative and uniform profit-seeking strategy focused on technology transfer with commercialization sought at the end of a R&D project [12,17,18]. However, today they are moving towards a strategy where they consider technology commercialization from the beginning of a R&D process and directly engage in commercialization [12,17,18].

Given this backdrop, in order to ramp up the diffusion of R&D results, it is necessary to reestablish the objectives of public research institutes and conduct R&D activities from a demand-oriented perspective. In other words, public research institutes need to have an empathy-based understanding of the demand side in order to identify latent problems and solve them through analytical and integrative thinking. To that end, the public sector needs to adopt the concept of Design Thinking, a demand-based problem-solving methodology that has been mainly used in the private sector. All in all, public sector R&D should ultimately depart from a strategy that merely focuses on technological innovation, and engage in organizational innovation based on a market-oriented mindset and actions.

This study shifts the focus to the public sector, specifically public research institutes, which play the primary roles in public R&D activities. It aims to look at how the adoption of Design Thinking as a methodology for technological innovation in the public R&D arena would affect technology performance, while suggesting implications for improving R&D efficiency and performance. Furthermore, given the influence that public R&D has on other public areas [4], Design Thinking could emerge a key driver of innovation across the public sector. Among the many areas in the public sector, we concentrate on the public R&D field, as output and outcomes related to scientific technologies based on public R&D activities have a significant impact not only on those having a demand for such R&D achievements but also on organizational capabilities, policies, and environments of the overall public sector [4].

### 2.4. R&D Performance in Public Sector

In general, R&D generally refers to creative and valuable knowledge that has been generated from a R&D process (R&D consist of creative and systematic work undertaken in order to increase the stock of knowledge—including knowledge of humankind, culture, and society—and to devise new applications of available knowledge [42]) and that can be openly used [43]. However, since different R&D projects have different objectives and characteristics, it is difficult to define R&D performance in a clear and uniform way [44].

As discussed in Section 2.3, studies on the private and public sectors use different performance indicators (refer to Table 3), as the two sectors have different R&D objectives. Private sector R&D professionals seek to secure product development capabilities in order to improve their competitiveness, accumulate knowledge, and foster a pool of talent for profit making. Given that the basic objective of the private sector lies in earning profits, studies on the private sector focus on financial performance-related indicators such as profits or sales. [24,29–32].

**Table 3.** Financial Performance Indicators in the Private Sector.

| Category | Performance Indicators | Reference |
|---|---|---|
| Financial performance | Sales | [24,29–32] |
| | Market share | |
| | Return on investment | |
| | Profits | |

Conversely, public R&D activities are aimed at preventing market failure in the science and technology field by sharing the risks of technology investment with the private sector and boosting private sector R&D investment for the public good [5,45]. In public sector R&D, markets for R&D outputs are almost non-existent, and the fact that public sector R&D consists of mainly exclusive suppliers requires it to have performance measurement indicators that are different from those for private R&D [45].

Public R&D performance can be assessed based on scientific, technology, economic, social, and infrastructure performances [46]. In the Act on the Performance Evaluation and Management of National Research and Development Projects, R&D performance means "scientific and technological performances, such as patents, theses, and standards that are generated through research and development projects, and other economic, social, and cultural performances that are either tangible or intangible." Unlike the other four performance indicators, technology performance represents a key indicator regardless of the type (R&D project types include basic research, short-term industrial technology development, mid/long-term industrial technology development, public technology development, regional research development, defense technology development, fostering of talent, facility establishment, performance diffusion, and international collaboration [47]) and period of an R&D project [44]. Technology performance refers to performances related to the development of technologies for direct or indirect industrial application. Major outputs and performance indicators include either tangible or intangible performances generated through R&D projects, such as intellectual properties including patents, products, and services [44]. The key objective of R&D is the development and possession of advanced scientific technologies by creating new knowledge, products, and technologies (developing and possessing advanced scientific technologies is important not only for pursuing sustainable growth, but also strengthening global competitiveness [46]) [43], while technology performance is a performance indicator that well reflects such an objective. Furthermore, new technologies generated through R&D are important factors that lead to improved economic performance [43]. Among the many different criteria for assessing technology performance, patent numbers are most frequently used (Representing a major output for R&D activities, patents guarantee exclusive rights of a patent holder. As such, they are regarded as a highly objective criterion for assessing technological and economic values of a developed technology [48–54], and have been thus widely used for evaluating the performance of public research institutes) [48,55–57].

In addition, we note that the scope of public institutes' R&D objectives is expanding beyond the contribution to industrial and economic growth through techno-logical innovation to include the improvement of people's quality of life by addressing social problems. Accordingly, long-term scientific effects or social impacts could serve as crucial indicators in measuring the performance of public sector R&D.

## 3. Apply Design Thinking in Public Sector

### 3.1. Research Question

Design Thinking has been regarded as an innovation methodology that generates innovative and sustainable strategies for strengthening competitiveness based on creative thinking [21]. Design Thinking offers business and public sector organizations a way of developing original products and services that meet latent demand [40]. Many precedent

studies argued that Design Thinking promotes innovations at companies when applied to existing business strategies and models [24,58–60]. A number of related studies have been released [23–29,58–60]. As an example, one study looked at the impacts of Design Thinking on innovation and performance, using a CEO of a Thai company listed on the Stock Exchange of Thailand (SET) [24]. From this, a company with design thinkers was shown to be more innovative [24]. That is, Design Thinking was found to have direct impact on innovation, affecting overall performance creation [21]. Another research looked at the relation between Design Thinking activities at a company's new product development team and level of innovativeness for the products [29]. The results demonstrated that the process of Design Thinking of the team boosted the usefulness and novelty of the products [29].

Aside from the aforementioned research on the product/service development at companies with design thinkers, there are also a number of studies on the role of Design Thinking in solving various problems that could arise in an organization [61–63]. In one study, employees at the Office of Human Resources Management and personnel of the Innovation Lab in the U.S. were offered a Design Thinking-related training program and were then interviewed for an assessment of the impact of the program. From this, Design Thinking was shown to be a useful methodology in U.S. public institutions since it encouraged the participation of stakeholders and improved decision-making processes [58].

The process encourages organizations to observe and interpret problems from a new perspective and pursue experimentation and risk with the discovery of explicit and potential needs of users from various perspectives, leading to innovative and creative solutions [10]. Accordingly, the adoption of Design Thinking should help public research institutes better respond to shifting R&D environments. That is, Design Thinking would likely pave the way for public research institutes to take on new roles, positively impacting their R&D performance. In fact, a number of public research institutes have introduced Design Thinking through means such as operating living labs or offering Design Thinking training programs for fostering talent pools [64].

Still, a substantial part of research for Design Thinking in the R&D arena has been concentrated on the private sector, analyzing its impacts on corporate performance. Furthermore, despite the differences between both private and public sector R&D, precedent studies often generalized that Design Thinking has a significant impact on public R&D performance based on research concerning private R&D. It is noteworthy that both private and public R&D differ from each other in terms of the key objectives for R&D activities and performance measurement. Private sector R&D professionals seek to secure product development capabilities in order to improve their competitiveness, accumulate knowledge, and foster talent for profit-making. Conversely, public R&D activities are aimed at preventing market failure in the science and technology field by sharing the risks of technology investment with the private sector and boosting private sector R&D investment for the public good [5,38]. In public sector R&D, markets for R&D outputs are almost non-existent, and the fact that public sector R&D consists of mainly exclusive suppliers requires it to have a performance measurement indicator that are different from those for private R&D [38].

Given that studies that have verified the effects of Design Thinking in the public sector have been scarce compared to those in the private sector, this study aims to conduct an empirical analysis on whether Design Thinking has any meaningful impact on public R&D performance and show that the impacts of Design Thinking could reach beyond public R&D to the overall public sector. Therefore, we define the following research questions and research model to validate our assumption.

Research Question 1. Will the application of Design Thinking, which has been thus far adopted mainly in the private sector, contribute to performance creation?

A program logic model is a systematic, visual way to understand and present the logical causal relationship with respect to the resources, activities, changes, and outcomes of a program. The model consists of four components, namely input, activities, outputs, and outcomes. In contrast to precedent studies that focused on the private sector, our study includes outcomes as a performance indicator, given that Design Thinking can be presented as an action and that public research institutes target long-term achievements—a reason why assessments of government policies or public research activities are performed often based on a logic model.

Research Question 2. Under the program logic model, would Design Thinking help public institutes not only achieve short-term technological results, but also fulfill their fundamental objective of contributing to solving social problems?

### 3.2. Research Model

In phase with precedent studies, this study regards activities to generate R&D performance as Design Thinking as a demand-based innovation methodology of public institutes. We have applied the steps presented in Table 2 to the elements of Design Thinking. Furthermore, based on R&D performance-related precedent studies and a program logic model, we categorize R&D performance into output (immediate, first-level results associated with a project) and outcome (the second-level results associated with a project) [63,64]. It appears as shown in Figure 1.

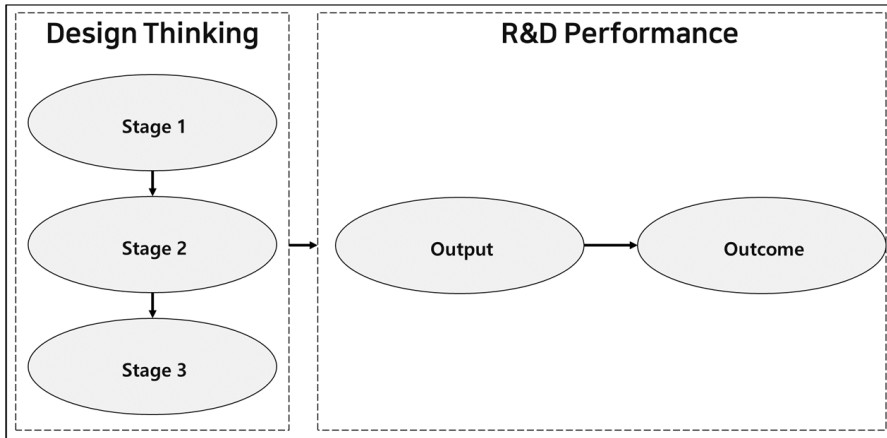

**Figure 1.** Research Model.

The details are as follows (Figure 1).

Stage 1 means understanding people by observing their motivation, emotion, senses, and reason. By taking a human-centered approach, design thinkers are able to discover explicit and potential needs of colleagues, clients, users, and customers (current and prospective) based on multiple perspectives. Through empathy, design thinkers observe the world in detail and use their insights to drive innovation. That is, it goes beyond simply having a market or user-oriented view to discovering the potential needs of users [32]. In a public sector R&D process of, empathy corresponds to efforts to understand consumers' needs. That is, it refers to conducting R&D based on an under-standing of users. An observation of a problem in relation to consumers' needs and opinions enables the discovery of various ideas that could solve a problem. Furthermore, in an R&D process, various ideas lead to the consideration of various factors that could affect R&D performance from an integrative approach. Based on human-centered observations, Stage 1 enables organizations to engage in a close relationship with consumers and discover their potential needs, which in turn paves the way for new, integrative thinking [65–71]. Furthermore, by putting oneself in other people's shoes and connecting with how they might feel about their problems,

it helps organizations to identify, reorganize, and materialize potential issues, actively devising various solutions and strengthening their integrative thinking skills [32,33,36].

Design thinkers with integrative thinking skills not only use analytical processes, but are also able to see all the contradictory aspects of a problem, which allows them to create new solutions (Stage 2) that improve on existing alternatives. This represents the essence of the ideation step. A vertical and rigid organizational structure [41] and performance indicator-focused R&D activities could undermine integrative thinking at public research institutes. However, considering studies arguing that both creativity and characteristics of public research organizations influence innovation activities, we believe that a market-oriented mindset could promote integrative thinking.

Based on considering various factors that could affect R&D results, integrative thinking should boost experimentalism that generates results by going beyond existing frameworks and thinking of new research methods and ideas. Looking at precedent studies, it is argued that existing and new concepts create an optimal alternative through an integrative thinking process (spanning exploration, elaboration, integration, and elimination). Thus, accounting for various factors in the course of the search for an ideal solution, integrative thinking should help reduce sunk costs, while coincidently boosting experimentalism (Stage 3) [32,72–74].

In pursuing experimentation and risks in the R&D process, researchers repeatedly test various ideas and accumulate knowledge and know-how, which in turn leads to improving R&D results. This process helps enhance economic efficiency by achieving cost and time savings and minimize a trial-and-error process [25,73,74]. That is, experiments to test various ideas based on an understanding of user needs should improve R&D results in terms of function, aesthetic impression, and value, while greater experimentalism is likely to result in a more successful R&D performance.

## 4. Methodology

### 4.1. Survey Methodology and Data

After designing the survey questionnaire, we analyzed Design Thinking and R&D performance of R&D personnel at public research institutes in Korea (The concept of R&D personnel entails those hired for conducting tasks directly related to R&D activities, and those who offer related services such as research and development managers, administrative workers, technicians, and office workers. [45]. R&D personnel refer to those who engage in one or more tasks of the following: scientific and technological tasks to meet a project's objective (i.e., engaging in experiments, research, and prototype production), R&D-related support tasks (i.e., planning and operating R&D projects, drafting reports, and managing computer networks, literature information, & document record), and other administrative support tasks with regards to R&D projects [43]. In order to study the impacts of Design Thinking on R&D performance at public research institutes, this research surveyed R&D personnel working in public research institutes.).

Public research institutes in Korea have played a pivotal role in national R&D as a key pillar in the national innovation system, and recently, the role of creating performance and increasing utilization through technology commercialization has been emphasized. This research is intended to verify what kind of performance the Design Thinking methodology creates within these institutes. As argued in previous studies, it is necessary to reduce the gap between the R&D stage of the public sector and the commercialization stage of the private sector for successful technology commercialization. This is because Design Thinking, which values empathy from a consumer's point of view, is believed to revitalize innovation in the public sector by reducing this gap.

The survey was conducted online for 17 days spanning July to August 2020. Respondents to this survey are mainly researchers working in government-funded and local government-funded research institutes among public research institutes and play a pivotal role in R&D in the public sector. In addition, among the R&D personnel surveyed for this study, those at colleges were researchers hired by colleges for conducting R&D tasks, excluding undergraduates, graduate students, and professors. While a college performs not only research but also education, this study excludes students and professors in order to better focus on the college's role as a research institute.

The number of respondents in this study tallied 348 people, including 56 with years in service of less than 5 years (16.0%), 84 with 5 to less than 10 years (24.1%), 72 with 10 to less than 15 years (20.7%), 49 with 15 to less than 20 years (14%), and 87 of 20 years or more (25.0%) (Table 4). This suggests that the sample offers a well-rounded pool of respondents in terms of years in service. For research model analysis, we performed SEM (structural equation modeling) using the STATA 15.1 program (Stata Corp LCC, College Station, TX, USA).

**Table 4.** Sample data.

|  |  | Number of Sample | Distribution Ratio |
|---|---|---|---|
| Organization | Universities and colleges | 131 | 38 |
|  | National research institute | 41 | 12 |
|  | Government-funded research institute | 107 | 31 |
|  | Local government-funded research institute | 69 | 20 |
| Job Group | Research Staff | 230 | 66 |
|  | Analytical Staff | 32 | 9 |
|  | Technical Staff | 49 | 14 |
|  | Misc. | 37 | 11 |
| Employment Period | Less than 5 years | 56 | 16 |
|  | 5 to less than 10 years | 84 | 24 |
|  | 10 to less than 15 years | 72 | 21 |
|  | 15 to less than 20 years | 49 | 14 |
|  | 20 years or more | 87 | 25 |

### 4.2. Operational Definition of Variables

According to the definitions in dictionaries and precedent studies, public research institutes in this study (the scope of public research institutes varies by study, as the types and tasks of public research institutes vary by country. Therefore, there is no specific definition of the scope of public research institutes, and the scope varies according to research. This study defined public research institutes as national research institutes, government-funded research institutes, local-government-funded research institutes, and universities and colleges based on previous studies [8,45]) entail universities and colleges, national research institutes, government-funded research institutes, and local-government-funded research institutes [5–8].

The operational definition for Design Thinking and R&D performance (in terms of output and outcome) has been established as seen in Table 5.

**Table 5.** Operational Definition of Variables.

| Variable | | Operational Definition | Related Studies |
|---|---|---|---|
| Design Thinking | Stage 1 | Conducting R&D based on an understanding of those on the demand side * and efforts to listen to their needs and views. | [29–31,39,75] |
| | Stage 2 | Efforts to solve problems in relation to the many factors that could affect R&D results from an integrative perspective | |
| | Stage 3 | Efforts to generate results by going beyond existing frameworks & thinking of new research methods & ideas | |
| R&D Performance | Output | Technology output created at the end of a R&D process, such as patents, prototypes, & new products | [75–78] |
| | Outcome | Impacts that a R&D result has, such as improved technological capabilities, reduced technological gap, & contribution to development of new technologies | [75–78] |

\* The managing institute of ongoing R&D tasks, or stakeholders in technology transfer & commercialization.

Output from R&D activities refers to first-level results, and includes research papers, reports, patents, processes, and information/knowledge. That is, it refers to qualitative and quantitative output created at the end of an R&D process [51,52]. Meanwhile, outcome refers to second-level results associated with an R&D project or impacts that an R&D result has on society and industry, such as the creation and exchange of knowledge, collaboration, formation of networks, reduced costs, improved revenue, and product innovation [77,78].

In this study, output or first-level results of R&D performance includes patents, prototypes, and products. Outcome (impact) or second-level results encompasses technology innovation and impacts on the development of other technologies. By analyzing the first and second-level results of R&D performance, we will demonstrate not only the impacts that Design Thinking has on R&D results but also its impacts on an industry and society.

## 5. Results

### 5.1. Factor and Reliability Analysis

Before hypothesis testing, we conducted a confirmatory factor analysis to review the validity of the variables used in our research model. Observed variables for the confirmatory factor analysis were Stage 1, Stage 2, Stage 3, Output, Outcome. We evaluated the results of the analysis by looking at fitness indices—the absolute fit index RMSEA (which tends to be relatively less affected by sample size) and incremental fit indices CFI and TLI. The TLI and CFI values came to 0.980 and 0.984, respectively, with both figures higher than the reference value of 0.9. Meanwhile, the RMSEA value arrived at 0.048, a level that is indicative of fit because it is lower than the reference value of 0.08.

The results for convergent validity analysis are indicated in Table 6. The average variance extracted (AVE) by construct ranged from 0.707 to 0.806, levels standing above the reference value of 0.6. Furthermore, composite reliability (CR) values ranged from 0.878 to 0.926, figures that exceed the reference value of 0.8, thereby indicating acceptable convergent validity in the measurement model. In addition, we measured internal consistency with Cronbach's alpha, whose values all surpassed 0.8 as an indication of good reliability.

### 5.2. Correlation

To analyze the correlation among major variables, we conducted a Pearson's correlation analysis. The correlation coefficients among the variables ranged between 0.1614 to 0.7532, as shown in Table 7. That said, the VIF values ranged from 1.06 to 3.11, remaining below 10, which indicates no multicollinearity. Furthermore, for discriminant validity verification, we looked at the average variance extracted (AVE) of constructs. With the AVE of constructs being greater than the square correlation between them, it was verified that the discriminant validity was secured.

**Table 6.** Confirmatory Factor Analysis of Major Variables & Results of Convergent Validity Analysis.

| Construct | | Metrics | Standardized Regression Weights | Average Variance Extracted (AVE) | Composite Reliability (CR) | Cronbach's α |
|---|---|---|---|---|---|---|
| Design Thinking | Stage 1 | E1 | 0.778 | 0.756 | 0.925 | 0.925 |
| | | E2 | 0.719 | | | |
| | | E3 | 0.796 | | | |
| | | E4 | 0.784 | | | |
| | Stage 2 | I1 | 0.612 | 0.766 | 0.908 | 0.906 |
| | | I2 | 0.645 | | | |
| | | I3 | 0.549 | | | |
| | Stage 3 | EX1 | 0.795 | 0.806 | 0.926 | 0.925 |
| | | EX2 | 0.825 | | | |
| | | EX3 | 0.766 | | | |
| R&D Performance | Output | TO1 | 0.774 | 0.707 | 0.878 | 0.866 |
| | | TO2 | 0.806 | | | |
| | | TO3 | 0.572 | | | |
| | Outcome | TI1 | 0.795 | 0.785 | 0.916 | 0.915 |
| | | TI2 | 0.833 | | | |
| | | TI3 | 0.753 | | | |

**Table 7.** Correlation & Discriminant Validity for Major Variables.

| | Stage 1 | Stage 2 | Stage 3 | Outcome | Output | Sqrt (AVE) |
|---|---|---|---|---|---|---|
| Stage 1 | 1 | | | | | 0.8695 |
| Stage 2 | 0.7248 *** | 1 | | | | 0.8752 |
| Stage 3 | 0.6131 *** | 0.7532 *** | 1 | | | 0.8978 |
| Output | 0.2464 *** | 0.1800 *** | 0.1614 *** | 1 | | 0.8408 |
| Outcome | 0.3367 *** | 0.2258 *** | 0.1996 *** | 0.6943 *** | 1 | 0.8860 |

*** $p < 0.01$.

*5.3. Path Analysis Results*

To measure model fit, we calculated fitness indices—the absolute fit index RMSEA (which tends to be relatively less affected by sample size) and incremental fit indices CFI and TLI. As a result, the TLI and CFI values came to 0.976 and 0.980, respectively, with both figures higher than the reference value of 0.9. Meanwhile, the RMSEA value arrived at 0.053, a level that is indicative of fit because it is lower than the reference value of 0.08. As such, the research model of this study was concluded to be suitable for hypothesis verification.

As shown in Table 8, Design Thinking as a demand-based innovation methodology of the public sector proved to have positive impacts on R&D performance.

The direct path from Stage 1 to Stage 2 showed a positive association ($\beta = 0.79$, $p < 0.01$). Coincidently, the direct path from Stage 2 to Stage 3 revealed a positive association ($\beta = 0.82$, $p < 0.01$). Lastly, the direct path from technology output to technology outcome suggested a positive association ($\beta = 0.74$, $p < 0.01$). That said, Stages 2 and 3 did not affect technology output as single elements.

**Table 8.** Path analysis.

| | Standardized Coefficients | Unstandardized Coefficients | Standard Error | Z-Value |
|---|---|---|---|---|
| Stage 1 → Stage 2 | 0.79 | 0.7671 | 0.050 | 15.27 *** |
| Stage 2 → Stage 3 | 0.82 | 0.8779 | 0.052 | 17.03 *** |
| Stage 1 → Output | 0.35 | 0.2919 | 0.087 | 3.34 *** |
| Stage 2 → Output | −0.11 | −0.0981 | 0.127 | −0.77 |
| Stage 3 → Output | 0.05 | 0.0411 | 0.090 | 0.46 |
| Output → Outcome | 0.74 | 0.7836 | 0.058 | 13.61 *** |
| $\chi^2$ = 185.783, df = 98, RMSE0.051, CFI = 0.982, TLI = 0.977, SRMR = 0.053 | | | | |

*** $p < 0.01$.

### 5.4. Verification of Indirect Effect

In addition, in this study, through the analysis of research model effects and understanding the direct, indirect, and total effects among variables (Table 9), we attempted to take a closer look at the structural relationships among the variables and verify the significance of mediating effects. Using bootstrapping, we verified the significance of direct, indirect, and total effects.

**Table 9.** Direct, Indirect, and Total Effect.

| | Direct Effect | Indirect Effect | Total Effect |
|---|---|---|---|
| Stage 1 → Stage 2 | 0.7671 *** | | 0.7671 *** |
| Stage 1 → Stage 3 | | 0.6734*** | 0.6734 *** |
| Stage 2 → Stage 3 | 0.8779 *** | | 0.8779 *** |
| Stage 1 → Output | 0.2919 *** | −0.0476 | 0.2443 *** |
| Stage 2 → Output | −0.0981 | 0.0361 | −0.0620 |
| Stage 3 → Output | 0.0411 | | 0.0411 |
| Stage 1 → Outcome | | 0.1915 *** | 0.1914 *** |
| Stage 2→ Outcome | | −0.0486 | −0.0486 |
| Stage 3 → Outcome | | 0.0322 | 0.0322 |
| Output → Outcome | 0.7836 *** | | 0.7836 *** |

*** $p < 0.01$.

We examined the dual mediating effects on the relationship between Stage 1 (which represents a mindset that empathizes with the demand side) and output and that between Stage 1 and outcome. The results revealed there was no zero between the lower limit values (0.060 & 0.761) and upper limit values (0.162 & 0.995). Therefore, it was concluded that Stage 2 and Stage 3 have dual mediating effects on the relationship between Stage1 and R&D performance (output/outcome). This gives us an implication that is the same as those in precedent studies—which suggested that a Design Thinking mindset could lead to Design Thinking actions. The results also imply that it is important to first understand the market before undertaking organizational innovation as an action method.

All in all, applying Design Thinking elements to public institutes' innovation process, starting with empathizing with the demand side, is anticipated to contribute to long-term R&D performance generation.

## 6. Discussion and Conclusions

### 6.1. Discussion

Helped by government-led technological innovation initiatives, Korea has enjoyed rapid technological growth to join the ranks of developed countries. That said, it has

been pointed out that a supplier-centered perspective has hindered the commercialization of R&D performance. As such, there have been rising calls for greater technology commercialization and a shift towards demand-focused R&D to ensure sustainable economic growth. Against this backdrop, this study conducted an empirical analysis, based on a view that Design Thinking as a demand-oriented innovation methodology could contribute to sustainable performance creation. Theoretical research on Design Thinking has been conducted for a long ago by now, and empirical studies on the contributions of Design Thinking in the private sector are being actively carried out. However, given that the public and private sectors have different R&D objectives and roles in the overall industry, using the results of precedent studies that focused on the private sector to forecast the performance of the public sector should yield only limited insights. With this in mind, we designed a research model based on consideration of the fact that public research institutes represent technology providers that seek to fill the technological gaps of private sector players and aim to resolve social problems over the long term. On this basis, we attempted to conduct an empirical analysis of the impacts of Design Thinking on R&D performance in the public sector to eventually demonstrate that Design Thinking would set the framework for advancing a demand-oriented perspective in the public sector.

The results of the study have shown that Design Thinking as a demand-based innovation methodology of the public sector positively impacted R&D performance through Stages 1 to 3.

Based on the research results, we present our discussion as follows.

Firstly, it was confirmed that researchers at public research institutes should clearly define the needs of related stakeholders before embarking on a R&D project, while considering their views throughout the whole R&D process. This is the first step as bridging the gap in the technology commercialization process [22]. In this research, it was revealed that public research institutes' efforts to understand the needs of those on the demand side in the R&D process made them more likely to conduct studies using integrative thinking skills, which in turn helped them solve problems from multiple perspectives. Notably, stakeholders should include both explicit and potential users. Given that Design Thinking at public research institutes could help find both explicit and potential user needs as well as promote networking and collaboration with various stakeholders, Design Thinking could serve as a strategic way for public research institutes to respond to the fourth-generation R&D model. In addition, Design Thinking would likely carry over to boosting R&D performance in relation to the fifth generation R&D model by promoting open innovation based on close partnerships with stakeholders

Secondly, a progression from Stages 1 to 3 should have a meaningful impact on technology outputs and outcomes. That is, developing a mindset presented in Stage 1 is a prerequisite for Stages 2 and 3 to lead to technology outputs and outcomes. and it helps reduce sunk costs while coincidently boosting experimentalism. This study proves that integrative thinking at public research institutes prompts researchers to take into account various factors that could affect R&D results while conducting R&D activities. As such, the practice facilitates the understanding of changing R&D environments and encourages new attempts and ideas.

Thirdly, it has been demonstrated that the impact of Design Thinking at public research institutes could go beyond just R&D results to the overall society and industry. As such, Design Thinking could lay the foundation for innovation in the public sector.

### 6.2. Conclusions

The results of the study have shown that Design Thinking as a demand-based innovation methodology of public sector positively impacted R&D performance through a process spanning empathy, integrative thinking, and experimentalism. Based on the research results, we present our conclusions as follows.

Firstly, empathy positively affected integrative thinking—a conclusion that remains in phase with the results of precedent studies that empathy based on observations from

multiple perspectives helps create multiple solutions and strengthen integrative thinking. In this research, it was revealed that public research institutes' efforts to understand the needs of those on the demand side in the R&D process made them more likely to conduct studies using integrative thinking skills, which in turn helped them solve problems from multiple perspectives.

Secondly, integrative thinking had positive effects on experimentalism. Notably, integrative thinking considers various alternatives before coming up with the optimal solution. This remains in phase with the results of precedent studies whereby integrative thinking helps reduce sunk costs while coincidently boosting experimentalism. This study proves that integrative thinking at public research institutes prompts researchers to take into account various factors that could affect R&D results while conducting R&D activities. As such, the practice facilitates the understanding of changing R&D environments and encourages new attempts and ideas.

Thirdly, experimentalism turned out to have a significant impact on R&D performance. This is similar to the result of precedent studies where repeated experiments helped find an optimal solution, while keeping the expended time and cost contained. This study verified that experimentalism has a meaningful impact on R&D performance at public research institutes, implying that the impact of Design Thinking at public research institutes could go beyond just R&D results to the overall society and industry. As such, Design Thinking would set the groundworks for innovations in the public sector.

Fourthly, empathy, followed by integrative thinking and experimentalism, was verified to have a meaningful impact on technology output and outcome. That is, an understanding of users' needs based on empathy has a significant impact on R&D performance when combined with integrative thinking (which understands contradictory aspects of a problem) and experimentalism (which pursues experimentation and risks).

Overall, R&D performance improves in phase with: (1) consideration of users' needs in the R&D process; (2) understanding of various factors in R&D environments; and (3) pursuit of experimentation and risks.

### 6.3. Implications and Limitations

Based on the results of this study, we have come up with the following academic and practical implications.

As for the study's academic implications, firstly, this research expanded the traditional scope of the study of Design Thinking (which was limited to companies in the private sector) to include the public sector, focusing on public research institutes of Korea, which play the primary roles in public R&D. Based on this, this study revealed that Design Thinking as a demand-based innovation methodology of public sector has a positive impact on the improvement of R&D performance at public research institutes. That is, this study expanded the scope of Design Thinking research and demonstrated that Design Thinking has significant impacts in both the private and public sectors, which should promote further discourses concerning Design Thinking going forward. Especially, having divided R&D performance into technology output and outcome, this study revealed that Design Thinking affects not only R&D results, but eventually overall society and industry. Therefore, it can be said that Design Thinking not only affects R&D performance but also could drive innovations in the public sector.

In short, this research suggests that Design Thinking is required for the improvement in R&D performance at public research institutes, and that it represents an innovation methodology that encourages such processes. Furthermore, it shows that the impacts of Design Thinking could reach beyond public R&D to the overall public sector.

The practical implications of this study are as follows.

Firstly, public research institutes should offer educational programs to help re-searchers improve their understanding of a wide range of areas, so as to consider various factors in conducting R&D projects. As Design Thinking emphasizes repeated experiments and uses feedback to supplement and improve R&D results, continuous demand-based R&D

performance measurement and improvement processes are necessary. For example, it is necessary to collect various customer experiences and opinions through customer satisfaction surveys, the Net Promoter Score (NPS), focus groups, and direct interviews, and improve R&D through feedback to undertake R&D tasks and system operation [79]. As a result, it promotes creative R&D experimentation that will likely ramp up the move towards a first-mover R&D strategy by increasing the job satisfaction of R&D personnel [80]. Furthermore, experts from various areas should engage in collaboration, so as to collect R&D data from multiple perspectives.

Secondly, clearly understanding the scope of users and seeking to reflect their needs in the early phase of a R&D process is a prerequisite for smooth progress of the whole R&D process that goes from planning to implementation to commercialization [3]. This is attributed to the fact that R&D results created based on a precise understanding of the demand side and stakeholders are more likely to be widely used and successfully commercialized. As such, public research institutes need to employ a demand-based strategy from a user-specific Design Thinking perspective in conducting R&D tasks. In addition, since impacts of R&D results at public research institutes could reach various fields such as the economy, society, and culture, public R&D results based on Design Thinking is predicted to set the groundworks for meaningful innovations across various fields in the public sector. That is, the introduction of Design Thinking at public research institutes should not only improve public R&D performance but also boost efficient operations in the public sector as well.

Even so, despite the many implications presented above, this study has its own limitations in that there is bias in data owing to the limited scope of survey materials. To present clearer causal relationships, the need has arisen to give more consideration on time or utilize panel data. Furthermore, owing to limitations in survey materials, this study narrowed its focus on technology performance. While it has been confirmed through precedent studies that technology performance represents a key indicator for assessment of R&D performance, the addition of other indicators (i.e., economic performance & social performance) in future studies will likely generate more findings.

As this study was conducted based on public research institutes in Korea, the discussion of this study can be further expanded by benchmarking comparable results derived from other countries. At this time, it will be meaningful to further study whether the difference in culture or norms between countries affects the relationship between Design Thinking and R&D performance [81], or even studying Design Thinking about whether there are elements other than norms, the results of which would enable expansion of the study.

**Author Contributions:** Conceptualization, Y.-w.S.; Data curation, S.L.; Formal analysis, S.L. and M.K.; Funding acquisition, Y.-w.S.; Investigation, S.L.; Methodology, S.L., M.K. and Y.-w.S.; Project administration, Y.-w.S.; Writing—original draft, S.L.; Writing—review & editing, M.K. and Y.-w.S. All authors have read and agreed to the published version of the manuscript.

**Funding:** This research was funded by Konkuk University in 2021.

**Informed Consent Statement:** Informed consent was obtained from all subjects involved in the study.

**Acknowledgments:** This paper was supported by Konkuk University in 2021.

**Conflicts of Interest:** The funders did not have any role in the design of the study, in the collection, analyses, & interpretation of data, in the writing of the manuscript, nor in the decision to publish the results.

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
