# Peer review of "Design Thinking for Public R&D: Focus on R&D Performance at Public Research Institutes"

_sustainability, doi:10.3390/su14137765_

Round 1

Reviewer 1 Report

Thank you for the opportunity to read your work.

I do agree that is important to study several contexts. The public sector is indeed less explored compared with private.

Innovation is a current issue. So, overall, the study seems to be relevant.

However, I have some concerns:

  • Some advices/~improvements
    • Please clarify in the text if your statements are about Korea or worldwide. For instance, line 67. The reference indicates KOrea but the text is not clear
    • Between line 78 and 86 the text has a different format.
    • It is not necessary to detail so much what design thinking is in Introduction section.
    • Theoretical background has 4 pages. It seems a bit to much for me
  • Main concerns
    • Your title, abstract and introduction clearly state that you aim to explore DT in public sector. That is ok. But, there is not a single Hypothesis considering the "public". Why not? This is an important part of your investigation. It should be considered as a variable to be tested.
    • It seems to me that your are assuming that since the respondents work in a public institution than your are automatically validating the "public" part. I'm not an expert in quantitative methods but I believe this is not the right way.
    • This is a serious issue in my opinion that compromises the entire study. And, or I´m actually wrong, or it is difficult to improve the study to be ready for publication. You would have to redo the entire research.
    • Section 3.1 makes 2 references to "public" and never related with the H design. Section 4 has 0 references to "public".
    • I know this question is difficult, but why are you trying to publish a study almost 2 years after the research?

Reviewer 2 Report

I see this study as interesting and important for public R&D activities management. However in my perception the DT is treated a little bit too formally, as we know many crucial elements of DT or even all of them, might be strongly used by a team even not being aware of that.

What I see as totally indispensable is to suplement discussion/conclusions section with managerial views on R&D activities in public institutions, in other words how to manage research teams to take more advantage of DT? Additionally, the sample description is not sufficient, are these respondents from one university? Good luck!

Reviewer 3 Report

In this study, the major drawback is the absence of any theoretical background. Authors are suggested to develop their hypotheses based on a solid relevant theory. Moreover, it will be highly appreciated if they could add a section related to the theoretical background.

Regarding hypotheses development, for each hypothesis, there is a need to write a separate justification paragraph. It will enhance the readability and understanding of your work to the layman. 

IT is good to split the long discussion into the separate section- Discussion, Conclusion, Implications and Limitations, and write clearly

Round 2

Reviewer 1 Report

Thank you for your improvements.

In my opinion the same issue remains.

You focus the your research motivation and problem in public sector but none of the hypothesis take the "public" into consideration. 
